# SKDream: Controllable Multi-view and 3D Generation with Arbitrary Skeletons

## Abstract

Controllable generation has achieved substantial progress in both 2D and 3D domains, yet current conditioning methods still face limitations in describing detailed shape structures. Skeletons can effectively represent and describe object anatomy and pose. Unfortunately, past studies are often limited to human skeletons. In this work, we generalize skeletal conditioned generation to arbitrary structures. First, we design a reliable mesh skeletonization pipeline to generate a large-scale mesh-skeleton paired dataset. Based on the dataset, a multi-view and 3D generation pipeline is built. We propose to represent 3D skeletons by Coordinate Color Encoding as 2D conditional images. A Skeletal Correlation Module is designed to extract global skeletal features for condition injection. After multi-view images are generation, 3D assets can be obtained by incorporating a large reconstruction model, followed with a UV texture refinement stage. As a result, our method achieves instant generation of multi-view and 3D contents which are aligned with given skeletons. The proposed techniques largely improve the object-skeleton alignment and generation quality. Project page at https://github.io/skdream3d. Dataset, code and models will be released in public.

## 1 Introduction

In view of representation dimension, 2D image generation (Ho et al., 2020; Song et al., 2020; Rombach et al., 2022), multi-view (2.5D) generation (Shi et al., 2023; Liu et al., 2023), and 3D generation (Hong et al., 2023; Li et al., 2023a; Xu et al., 2024) have been promoted and made great progress successively. To realize more flexible and controllable generation, conditions beyond text have drawn considerable attention. 2D image conditions (*e.g.,* edge maps, human skeletons, and concept references) (Zhang et al., 2023; Ruiz et al., 2023) have been well studied. Similarly in 3D generation, analogous 2D conditions have also been studied (Li et al., 2023d). Additionally, 3D conditions like simple shapes (Dong et al., 2024) have also been explored.

Although the aforementioned conditions in controllable generation complement text descriptions, they still struggle in precisely describing shape structures. In contrast, skeletons, among various types of conditions, exhibit superior ability to depict shape structures: *(i) Representation of object anatomy.* A skeleton can efficiently represent various 3D structures with sparse joints and bones. It would be cumbersome for other conditions to represent anatomy. *(ii) Articulation into different poses.* Skeletons are widely used for character animation in computer graphics (Kavan et al., 2007; Baran & Popović, 2007) due to their simplicity and efficiency. Other conditions such as rough shapes (Dong et al., 2024) are inconvenient to deform into different poses. *(iii) Freedom of editing.* Given an initial skeleton, users can freely add new structures or modify joint positions and bone sizes to create their ideal shapes. Examples for demonstration are in Fig. 1.

Despite these advantages, previous studies (Zhang et al., 2023; Mou et al., 2024; Ju et al., 2023; Zhang et al., 2024b; Huang et al., 2024) on skeletal conditioned generation are limited to human skeletons. From the perspective of generalization, we would like to ask: *Is it possible to use arbitrary skeletons as conditions to generate any creatures or even general objects?*

Towards this aim, we believe that two main issues hinder the use of arbitrary skeletal conditions for generation: *(i) Lack of large-scale object-skeleton pairs for training*. Extensive studies (Cao et al., 2017; Fang et al., 2022; Martinez et al., 2017) on 2D/3D human pose estimation make human-skeleton paired data easy to obtain. However, when skeletal structures are unknown, estimating

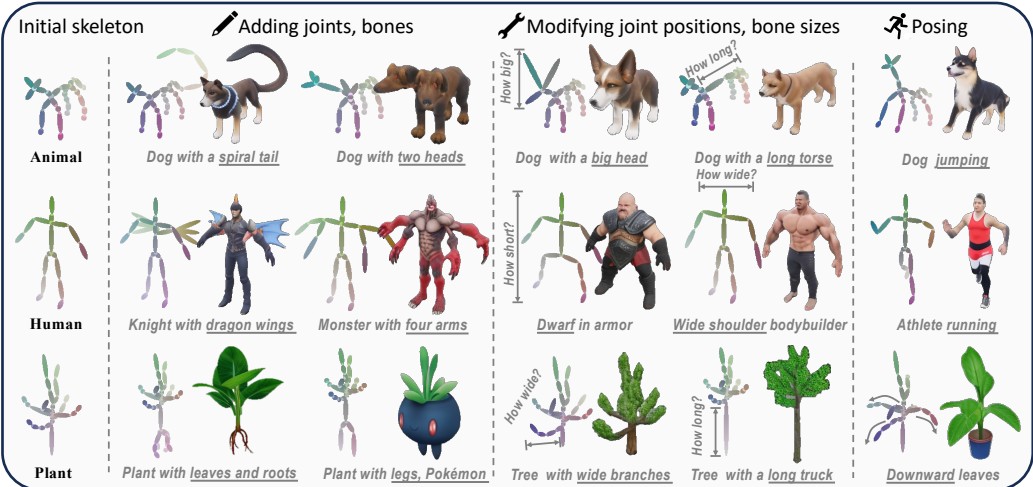

(a) Text and skeletons for **Collaborative Appearance and Shape Controlling**

(b) Skeleton-based editing for **Accurate Anatomy and Pose Controlling**

Figure 1: **Demonstration of skeletal conditions for controllable generation.** We argue that ***skeletons and text provide complimentary description for shape and appearance respectively***, as shown in (a). Moreover, ***flexible and accurate controlling of object anatomy and pose can be realized by editing the joints and bones in skeletons***, as shown in (b). Arbitrary skeletal structures are supported in our framework. Multiple views are generated and only front view images are shown.

arbitrary skeletons from 2D images or videos becomes challenging due to its ill-posedness. ***(ii) Insufficiency of 2D information to describe arbitrary skeletons***. Human skeletons are simple and can be described by a fixed set of 2D joints. However, complex skeletons suffer from self-occlusion and ambiguity, which necessitates 3D information to fully determine their anatomy and pose.

To address these challenges, we focus on multi-view and 3D generation with skeletal conditions. For data scarcity problem, we construct ***a large-scale dataset Objaverse-SK containing mesh-skeleton pairs.*** Textured meshes are selected from Objaverse (Deitke et al., 2023) by semantic classes to form a subset. In order to realize reliable mesh skeletonization, we propose a new pipeline to generate skeletons with sparse joints from meshes. The pipeline mainly consists of curve skeleton extraction and curve simplification. Our pipeline achieves 80% success rate, largely outperforming previous deep learning based method RigNet (Xu et al., 2020) (15% success rate).

To fully control object anatomy and pose, we build the ***skeletal conditioned generation model in a multi-view manner.*** We represent a 3D skeleton with conditional skeleton images by *Coordinate Color Encoding (CCE)* to reduce ambiguity. Joints and bones are encoded with unique colors according to their 3D positions. For condition injection, we designed a *Skeletal Correlation Module (SCM)* to extract features from these conditional images and then generate multi-view images for the object. Later, a Large Reconstruction Models (LRM) is employed to produce 3D assets from the multi-view images. To address potential blurriness due to the low-resolution inputs and reconstruction inaccuracies, we enhance appearance quality using a texture refinement stage that up-samples the multi-view images to higher resolutions and refines the original texture in UV space.

The experimental results indicate that our framework achieves instant generation of multi-view and 3D contents which are aligned with given skeletons. The proposed coordinate color encoding and

the skeletal correlation module significantly improve the object-skeleton alignment score, and accelerates model convergence by $5\times$. 3D assets conforming to the given skeleton can be generated in $\sim$20s and refined in $\sim$60s. To the best of our knowledge, this work is a pioneer in achieving arbitrary skeletal conditioned generation with following contribution:

- Constructing **the first large-scale dataset**, Objaverse-SK, containing mesh and skeleton pairs that cover diverse skeletal structures. We developed a new pipeline for generating sparse skeletons from meshes with a high success rate.
- Proposing a **multi-view and 3D generation pipeline** for arbitrary skeletal conditions. This includes *coordinate color encoding* for compact condition representation and the *skeletal correlation module* for effective condition injection.

## 2 RELATED WORK

**Controllable 2D Generation.** Based on image diffusion models like Stable Diffusion (Rombach et al., 2022), versatile controlling conditions have been studied. In terms of spatial controlling, ControlNet (Zhang et al., 2023) and other similar works (Mou et al., 2024; Zhao et al., 2024) train a side network for spatial conditions such as edge maps, normal maps and human skeletons. Some works focus on human image generation from skeletons (Ju et al., 2023; Wang et al., 2024a; Hu, 2024). Box-based instance controlling is also concerned in some works (Zheng et al., 2023; Li et al., 2023c; Zhou et al., 2024). As for content controlling, (Ruiz et al., 2023) finetunes the model to bind the given subject with an identifier in text prompt. (Ye et al., 2023; Chen et al., 2024b) train an adapter to inject styles or concepts to the model. Some works (Liang et al., 2024; Wang et al., 2024b; Li et al., 2024) also focus on human ID control. Besides, some methods (Meng et al., 2021; Bansal et al., 2023; Mo et al., 2024; Ohanyan et al., 2024) can achieve conditional generation without additional modules or fine-tuning.

**Controllable 3D Generation.** Content controlling in 3D generation can be easily realized by image-to-3D generation, which has been studied by plenty of works (Hong et al., 2023; Tang et al., 2024; Li et al., 2023a; Xu et al., 2024). However, in the image-to-3D paradigm, spatial controlling for 3D generation is not as easy as content controlling. Coin3D (Dong et al., 2024) presents a framework to control the multi-view diffusion and 3D generation by shape proxies, i.e. combination of simple basic shapes. Sculpt3D (Chen et al., 2024a) enhances text-to-3D generation with retrieved 3D priors. Sherpa3D (Liu et al., 2024) proposes to generate a coarse shape with a 3D diffusion model and refine the shape with SDS (Poole et al., 2022). Clay (Zhang et al., 2024a) designs a transformer-based (Vaswani, 2017; Peebles & Xie, 2023) 3D diffusion framework and various conditions like images and point clouds can be injected through cross-attention layers. Some works for 3D human or avatar generation (Liao et al., 2024; Huang et al., 2024; Zhang et al., 2024b) uses human skeleton as the condition in 2D or 3D space. A recent work (Li et al., 2023d) realizes 3D generation with single-view 2D spatial conditions like normal maps and edge maps by conditional multi-view generation and 3D reconstruction. Our work shares the similar workflow, but we focus on general skeleton conditioned generation, which has never been studied by previous works.

**Mesh Skeletonization.** Various algorithms were designed for extracting skeletons from 3D meshes. (Tagliasacchi et al., 2012) and (Bærentzen & Rotenberg, 2021) compute curve skeletons (C-S) via iterative mesh contraction operations. (Dou et al., 2022; Wang et al., 2024c) proposed to extract skeletons medial axis transformation skeleton (MAT-S) by point selection and connection prediction. C-S and MAT-S can serve as shape representation, while human-made skeletons (H-S) are often different from them. Since the main purpose is animation, H-S only contain sparse joints and bones. Some works (Xu et al., 2019b; 2020) propose data-driven approaches to learn mesh skeletonization from human annotated data. In this work, we have tried learning-based method (Xu et al., 2020) but found the results were not satisfactory. Therefore, we develop a new pipeline to generate skeletons which are as sparse as H-S while keep the shape of C-S.

## 3 DATASET CONSTRUCTION

### 3.1 DATA PREPARATION

The largest existing open dataset containing mesh-skeleton pairs is ModelResources (Xu et al., 2019b). There are around 3,000 3D meshes without textures. The scale is not sufficient to train

Figure 2: **Illustration of the pipeline for mesh-skeleton pair generation** (§3.2). Curve skeleton is first extracted from the given mesh, followed by simplification of parted curves. The curve graph is converted to a tree as final skeleton.

a text-driven generative model, and it lacks textures for appearance modeling. To address these limitations, we construct a dataset with $8\times$ larger scale with color textures. Our dataset, named Objaverse-SK, is built upon a large-scale 3D dataset Objaverse (Deitke et al., 2023). Although our data generation pipeline is applicable to a broad range of object categories, we focus on three main categories including "Animals", "Human Shapes" and "Plants", as they can typically be represented by tree-structured skeletons. Category labels are obtained from G-Objaverse (Qiu et al., 2024). Consequently, our dataset contains 24,000 3D meshes, consisting of 15,000 animals, 6,000 human shapes and 3,000 plants. Text prompts of these models are generated by Cap3D (Luo et al., 2024).

### 3.2 SKELETON GENERATION

In order to obtain mesh-skeleton pairs, a proper method for generating skeletons from meshes is crucial. There are two concerns: the skeleton structure and success rate. The skeleton structure should properly describe the object anatomy and be able to used for posing. Moreover, an ideal method should generate reasonable skeleton structures with a high success rate. We tested a learning-based method RigNet (Xu et al., 2020). Although the generated skeleton structures can be close to human annotations in its training data, it tends to be unstable on diverse anatomies and it only produces symmetric skeletons (results are in Sec. 5.1).

**Skeleton extraction.** To enhance flexibility and robustness, we design a new reliable pipeline, utilizing curve skeletons as the intermediate representation. Illustration of the pipeline is in Fig. 2. Considering the structural inconsistency between curve skeletons and human-made skeletons, we further convert dense curves into sparse joints and bones. The detailed pipeline is elaborated below. 1) Initially, Mean Curvature Flow (MCF) (Tagliasacchi et al., 2012) is employed to generate curve skeletons from meshes robustly. 2) Next, we build a graph from the set of curves generated from the mesh, consisting of dense nodes and edges. Intersection nodes (degree>1) are recognized and the graph is divided into several parts by these nodes. 3) In each part, the curve does not contain any branch so it can be simplified by Douglas-Peucker algorithm (DP) (Douglas & Peucker, 1973) into line segments with fewer points.

**Tree conversion.** At this stage, the basic shape of the skeleton is established, but the root position and the bone direction between joints still need to be determined. The problem can be regarded as graph to tree conversion. First, a spanning tree is build from the graph to eliminate cycles. We then identify high-degree intersection nodes as candidates for the root. To ensure an efficient structure, the skeleton is configured by selecting the tree with the minimum height among these candidates. This approach ensures that the root node is located at a significant intersection, minimizing the distances between the root and other joints. More details of the full pipeline can be found in appendix.

## 4 METHOD

### 4.1 SKELETAL CONDITIONED MULTI-VIEW GENERATION

As the dataset is constructed, we consider to build the conditional generative model based on it. Since unconditional multi-view diffusion models have been well studied, we directly start from a base model MVDream (Shi et al., 2023) and focus on the conditional generation. Mainly two problems are concerned: how the skeleton is represented and how it is injected into the model.

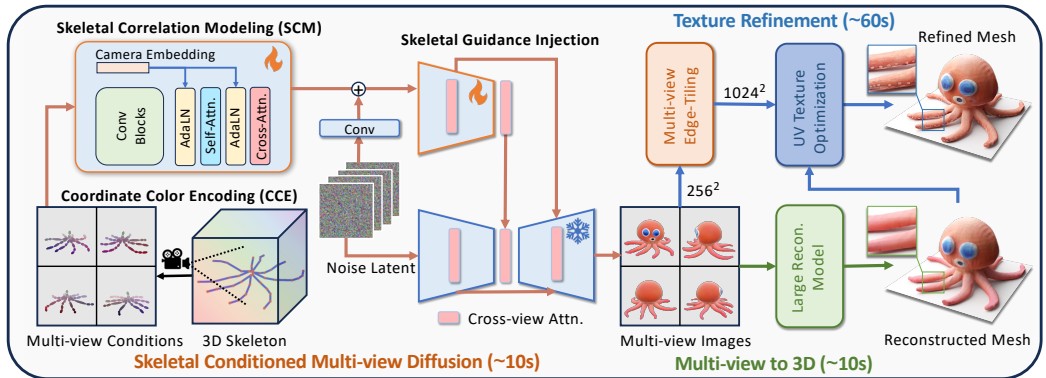

Figure 3: **Illustration of the pipeline for skeletal conditioned multi-view and 3D generation** (§4). The 3D skeleton is projected into 2D images and represented by coordinate color encoding. The correlation of skeletal images are extracted by skeletal correlation module, and then fused with the noise latent. Multi-view images are first generated and then 3D textured mesh is reconstructed. The texture is further refined via UV-space optimization.

**Skeletal condition representation.** As we want to generate images which aligns with the given skeletons, using spatial guidance for the diffusion model is a reasonable way. Similar to 3D meshes, skeletons can also be projected onto image planes as 2D conditions. However, depth information is lost during the projection, posing a significant challenge for spatial guidance. Unlike meshes, skeletons only consist of joints and bones, which can easily cause both semantic and spatial ambiguity as illustrated in Fig. 8. Thus, incorporating richer information is crucial to mitigate such ambiguities.

**Coordinate Color Encoding (CCE).** In order to preserve 3D information, we encode joint coordinates using spatial colors. While prior works (Li et al., 2023b; Wang et al., 2019) use canonical color map for shape representation, our approach focuses on representing skeletons with sparse joints and bones. We begin by normalizing skeletons within a canonical cube $[0, 1]^3$. Each position in this cube corresponds to a unique color, with RGB values precisely matching the positional coordinates. As a result, the 2D conditional image can represent the 3D spatial positions of the skeleton. For bones, we assign the color based on their midpoint. Additionally, we incorporate normalized values of view-dependent inverse depth of the skeleton as the alpha channel (CCE-D). With the absolute spatial coordinates and relative depth encoded in the conditional images, there will be more precise and richer guidance information for generation.

**Skeletal condition injection.** Spatial conditions like canny edges and normal maps have been investigated in 2D image diffusion models. In ControlNet (Zhang et al., 2023), the conditional image is encoded by convolution blocks, resulting in an output spatial size that matches the latent size. Then, the condition features are added to the latent features. The encoder of the original diffusion model is copied as a side network to produce guidance features, which are fused with the original features in the decoder. Our pipeline adopts this paradigm from ControlNet, and we further enhance it with a more effective condition feature extraction module.

**Skeletal Correlation Modeling (SCM).** For a skeleton in 3D space, we first project it into multi-view images as 2D conditions. Given the sparse nature of skeletal conditions in the spatial dimension, convolution blocks lack global modeling capacity. To address this, we design a Skeletal Correlation Module (SCM) to enhance the condition features by modeling the anatomy correlation among different parts of a skeleton, and the view correlation for different projection views. The structure of the module is in Fig. 3. *(i) First, anatomy correlation is extracted by a self-attention layer*, which constructs the global skeleton features for each view. *(ii) Then, the cross-view correlation is modeled by a cross-attention layer*, allowing the extraction of correspondences among skeleton images from multiple views. This enables the model to recognize identical joints in different views. In addition, we use adaptive layer normalization (Xu et al., 2019a) to fuse the camera pose embedding with the skeletal features. Associating each skeleton image with a camera pose aids in generating view-dependent object shapes. Although similar attention layers exist in the U-Net, adding correlation modeling layers during condition encoding significantly accelerates model training, achieving convergence 5 times faster (Fig. 10).

## 4.2 MULTI-VIEW IMAGES TO 3D GENERATION

**Instant reconstruction.** Given the generated multi-view images, we use a Large Reconstruction Model (LRM), specifically InstantMesh (Xu et al., 2024) for fast textured mesh reconstruction. However, the reconstructed textures often appear blurry. On the one hand, the resolution of generated images is $256^2$, which struggles in capture fine details. On the other hand, the appearance quality also degrades during reconstruction. In order to recover and further enrich the appearance, we introduce a new refinement stage.

**Appearance refinement.** First, the generated multi-view images are up-scaled 4 times into $1024^2$ by Stable Diffusion with ControlNet-Tile (Zhang et al., 2023). In order to keep multi-view appearance consistency, we perform view-concatenated inference, allowing attention layers to be shared by multiple views in a training-free manner. In addition, ControlNet-Edge (Zhang et al., 2023) is used to maintain the shape consistency during tiling. Canny edges (Ding & Goshtasby, 2001) are extracted as the additional condition. Once tiled, the high resolution images are used to refine the reconstructed texture. A learnable 2D texture $u$ in UV space is created and initialized as the reconstructed texture $u_0$, and then images are rendered through differentiable rendering for given camera views $c_i$. The MSE loss is optimized between the rendered images and tiled high-res images. Moreover, a regularization term is added to maintain consistency in UV space:

$$\mathcal{L}_u = \sum_i ||I_i^h - \mathcal{R}(u, c_i)||_2^2 + \lambda * ||u - u_0||_2^2. \tag{1}$$

$\mathcal{R}(u, c_i)$ is the image rendered from the mesh by differentiable rendering, and $I_i^h$ is the corresponding high-res image. Since the high-res images can not cover every position on the mesh, some regions of $x$ will not be optimized, e.g. bottom of the object. We found these regions are not stable during optimization and may produce unexpected artifacts (see Fig. 11). The regularization term will help the optimized texture maintain the appearance from $u_0$ in these regions. Consequently, the high-frequency details can be learned in covered regions while the global consistency can also be achieved in uncovered regions. The optimization could be finished within 15 seconds.

## 4.3 OBJECT-SKELETON ALIGNMENT EVALUATION

**Contrastive alignment.** In order to measure how much an object is aligned with a skeleton, we develop a new evaluator, named as Contrastive Object-Skeleton Alignment (COSA). We use the self-supervised DINOv2 (Oquab et al., 2023) as the backbone $F$ to extract both object and skeleton features. Then, the alignment adapter $G_\theta$ consisting of several self-attention layers is used to modulate the features. The adapter ends with a average pooling layer to aggregate the aligned features into a vector. Similar to CLIP (Radford et al., 2021), we train the adapter with contrastive learning by InfoNCE loss (Sohn, 2016; Oord et al., 2018). Finally, the skeleton alignment score (SKA) can be calculated by cosine similarity between the features from an object image $x$ and a skeleton image $y$ as $\mathcal{S}_{\text{SKA}}(x, y) = \cos(G(F(x)), G(F(y)))$.

**COSA guided diffusion.** Based on COSA, another conditional generation pipeline can also be realized, following the approach proposed in (Bansal et al., 2023). On each denoising time step $t$, the approximate clean image $\hat{x}_0$ is estimated from the predicted noise $\epsilon_t$ as in DDIM (Song et al., 2020). The estimated clean image and skeleton condition are fed into COSA to calculate the alignment loss $\mathcal{L}_{\text{COSA}}(\hat{x}_0, y) = 1 - \mathcal{S}_{\text{SKA}}(\hat{x}_0, y)$. Then the predicted noise is modified by the gradient of the alignment loss for actual denoising:

$$\hat{\epsilon}_t = \epsilon_t + s(t) \cdot \nabla \mathcal{L}_{\text{COSA}}(\hat{x}_0, y) \tag{2}$$

where $s(t)$ controls the guidance strength. With the additional guidance of the alignment loss, the generated object will tend to follow the conditional skeleton $y$.

## 5 EXPERIMENTS

### 5.1 RESULTS OF MESH SKELETONIZATION

We compare our method with a learning-based method RigNet (Xu et al., 2020), and the results are shown in Fig. 4. RigNet only produces symmetric skeletons so the flexibility is limited, resulting in a success rate around 15%. On the contrary, our method runs without limitation of symmetry and achieves better joint/bone alignment. It produces more reliable results with 80% success rate. More details and results can be found in appendix.

| Method\SKA Score | Training | Mean$_{Inst.}$ | Mean$_{Class}$ | Animals | Humans | Plants | Apodes | Bipeds | Quadrupeds | Arthropods | Wings |
|---|---|---|---|---|---|---|---|---|---|---|---|
| SDEdit (Meng et al., 2021) | ○ | 48.90 | 45.51 | 54.81 | 47.33 | 35.40 | 50.17 | 60.76 | 57.45 | 37.83 | 52.27 |
| SDEdit+COSAG | ◗ | 51.80 | 47.82 | 58.91 | 42.57 | 41.99 | 53.53 | 62.81 | 62.22 | 50.00 | 59.84 |
| **Ours** | ● | **81.13** | **72.63** | **92.90** | **80.19** | **44.80** | **93.92** | **88.56** | **95.98** | **91.45** | **93.76** |

Table 1: **Quantitative comparison of object-skeleton alignment (SKA) score** (§5.2). Alignment scores are calculated over three classes (blue) and five sub-classes of animal (green). The average score over all instances and three classes (pink) are also shown. Highest scores among all methods are bold and highest score among baseline methods are underlined.

| Method | Training | PickScore | | | | CLIP Score | | | |
|---|---|---|---|---|---|---|---|---|---|
| | | Win Rate | Animals | Humans | Plants | Mean$_{Inst.}$ | Animals | Humans | Plants |
| SDEdit (Meng et al., 2021) | ○ | 23.10 | 22.85 | 22.77 | 24.37 | 26.87 | 27.09 | 27.76 | **24.94** |
| SDEdit+COSAG | ◗ | 25.78 | 24.42 | 27.68 | 27.50 | 26.65 | 27.11 | 27.35 | 24.18 |
| **Ours** | ● | **51.12** | **52.73** | **49.55** | **48.13** | **27.51** | **28.10** | **28.24** | 24.63 |

Table 2: **Quantitative comparison of PickScore and CLIP Score** (§5.2). Scores are calculated over three classes (blue) and averaged over all instances (red). Highest scores among all methods are bold and highest score among baseline methods are underlined.

## 5.2 RESULTS OF MULTI-VIEW GENERATION

**Evaluation protocols.** We select 56 skeletons from our dataset for evaluation. The evaluation set covers three main classes: animals, human and plants. As animals include diverse skeleton structures, we further divide it into more detailed subclasses (examples are shown behind): Apodes (fish, snakes), Bipeds (ducks, penguins), Quadrupeds (dogs, bears), Arthropods (scorpions, crabs), Wings (birds, dragons). Three evaluation metrics are considered for multi-view generation: SKA Score for skeletal alignment, PickScore for image quality and CLIP Score for textual alignment. The evaluation results of other categories in ShapeNet (Chang et al., 2015) are provided in appendix.

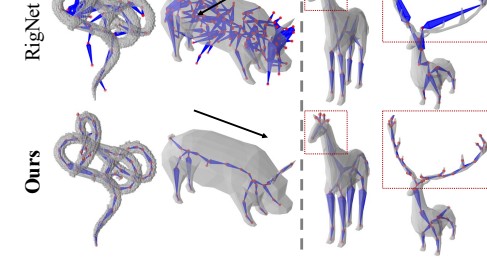

Figure 4: **Comparison of skeletons generated** from 3D meshes by RigNet (Xu et al., 2020) and our method (§5.1).

**Baseline methods.** Since there is no prior work that can achieve arbitrary skeletal conditioned generation, we implement two methods for comparison. The first baseline is SDEdit (Meng et al., 2021). The process starts from condition images, followed by adding noise on them with a time step (set as 0.7). Then clean images are generated by denoising steps. The method is totally unsupervised. The second baseline is the COSA Guidance (COSAG) derived from (Bansal et al., 2023), which is elaborated in Section 4.3. The guidance strength is set as $s(t) = 7.5\sqrt{1 - \alpha_t}$. Since we found it can not achieve stable results, it is combined with SDEdit. The method requires an extra model so it is partially supervised. Ours is fully supervised on object-skeleton pairs.

**Qualitative comparison.** The qualitative results are shown in Fig. 5. Given skeleton images in four views as condition, SDEdit can produce images following the skeleton. However, limited by the editing capacity, the generated objects often have wrong anatomy. For example, the snake body is apart, and the donkey body is generated as wood. When it is enhanced by the COSAG, the quality of generated contents is improved in some cases but still not satisfactory. Compared with them, our results show superior quality and are more consistent with both the skeletal and textual conditions.

**Quantitative comparison.** Comparison results of skeleton alignment are shown in Table 1. Training-free methods have around 50 SKA scores, while ours is around 80. Among three classes, animals tend to have higher alignment scores while plants have lower scores. Since the plants may have more complex structures and sometimes are hard to be represented by skeletons. For five sub-classes, our method achieves constantly higher alignment scores.

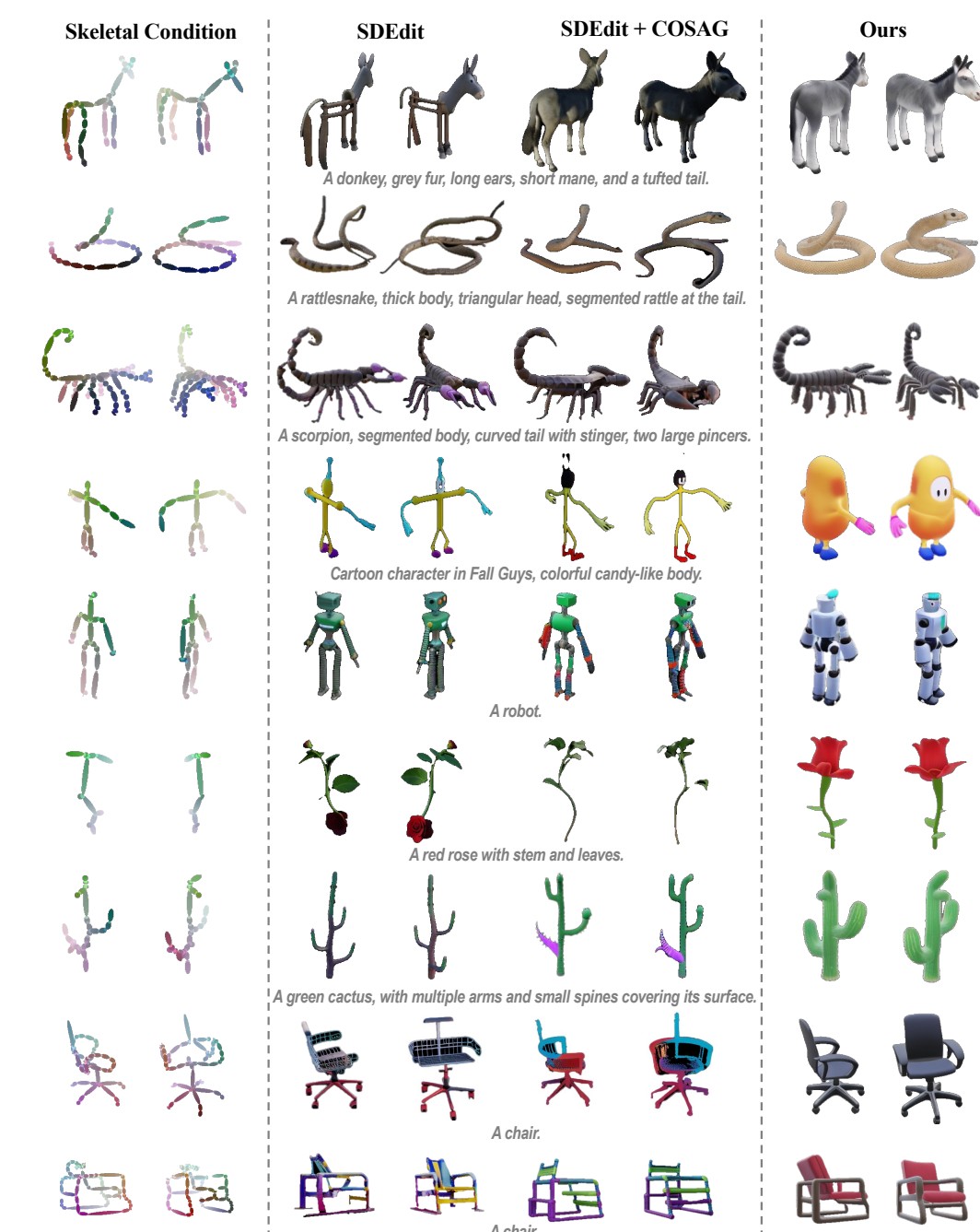

Figure 5: **Qualitative comparison of skeletal conditioned multi-view generation** (§5.2). Conditional skeletons are shown in left. Four views are generated and two views are shown for simplicity.

## 5.3 RESULTS OF 3D GENERATION

**Texture refinement.** Results of 3D reconstruction from multi-view images are shown in Fig. 6. The raw reconstructed results and refined results are compared. The raw textures are blurry and lack details, while the proposed refinement stage can significantly enhance the texture quality.

**Rigging and animation.** Since our framework can generate the textured mesh aligned with a given skeleton, the mesh can directly be rigged for animation, as shown in Fig. 7. The motion sequence of a skeleton could be made by artists or captured by Mocap methods. The mesh can be generated in an aligned manner at the canonical pose by our method, and then rigged and skinned for animation.

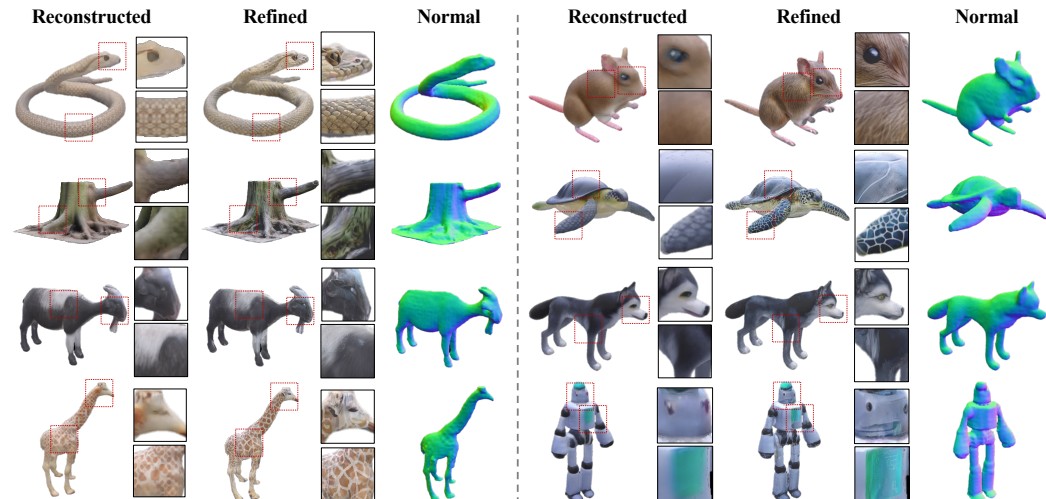

Figure 6: **Comparison of textured meshes before and after refinement** (§5.3). Rendered color and normal images are shown. Local areas are enlarged for better viewing.

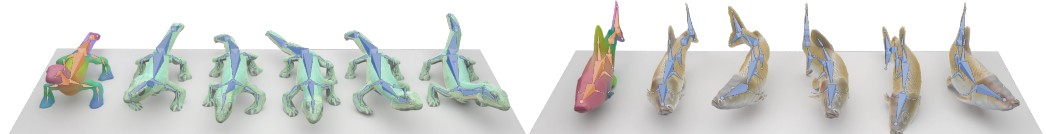

Figure 7: **Demonstration of mesh rigging and animation** (§5.3). The textured mesh are generated from the skeleton (orange). Since they are aligned, the mesh can be directly articulated by the skeleton (blue). Skinning weights of different joints/bones are visualized by different colors (left).

## 6 ABLATION STUDY

### 6.1 SKELETAL CONDITION REPRESENTATION

The skeletal condition representation we use consists of coordinate color encoding (CCE) with depth alpha (D). The ablation results are shown in Fig. 8 and Fig. 9. Richer information in conditions can help the model to determine the content better. As a result, higher image quality can be achieved. In Fig. 8 right, the skeleton of a penguin is highly ambiguous. If CCE-D is used, the body pose and orientation of the penguin can be successfully inferred from colors. From Fig. 9, the quantitative results indicate that CCE-D brings greater improvement for complex skeletons of animals and plants than simple skeletons of human shapes.

### 6.2 SKELETAL CORRELATION MODELING

With richer information encoded in the condition, how to extract features from the condition also counts. The corresponding module in previous works (Zhang et al., 2023; Li et al., 2023d) mainly consists of convolution blocks. Different from them, since multi-view condition of sparse skeletons is used in our setting, correlation modeling needs to be considered. We show the effect of skeletal correlation module in Fig. 10. SCM with layer normalization (LN) achieves $4\times$ faster convergence speed, compared with original convolution blocks. Furthermore, if LN is replaced with the adaptive LN (AdaLN), convergence can be further accelerated. The model can achieve a SKA score of 75 within 1k training steps. The results indicate that for spatial guidance, extracting global features from conditional images are crucial for conditional learning.

### 6.3 3D APPEARANCE REFINEMENT

We show the ablation results of appearance refinement in Fig. 11. The refined appearance contains rich and clear details such as snake scales and wood grain, compared with the reconstructed results. However, artifacts also appear in the regions which are not covered by high-res images. With the

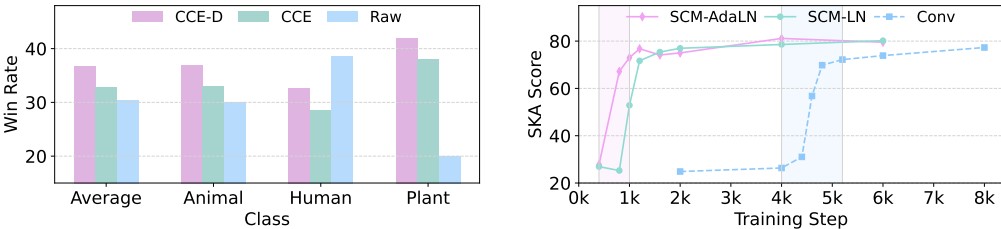

Figure 8: **Ablation study of coordinate color encoding with depth alpha (CCE-D)** (§6.1). Multi-view images are generated by different models with different condition types.

Figure 9: Comparison of PickScore among different skeletal representation types (§6.1). CCE with depth (CCE-D) achieves higher win rate.

Figure 10: Comparison of SKA Score among different conditional modules (§6.2). SCM with AdaLN achieves 5x faster convergence.

help of UV space regularization, the artifacts are effectively removed in uncovered regions. As a result, natural and consistent colors are maintained from original textures during optimization.

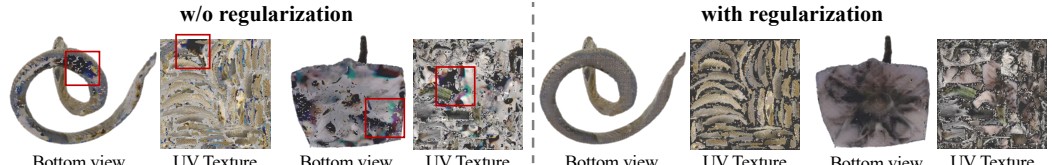

Figure 11: **Ablation study of UV space regularization** (§6.3). Bottom views and UV textures are shown. Front views of the snake and the tree stump can be found in the first column of Fig. 6.

## 7 LIMITATION AND FUTURE WORK

Since our work is the first one achieving arbitrary skeletal conditioned generation, there are still many problems can be further studied. The skeletons we currently use may have limited description ability for non-tree structured objects. More general yet efficient shape representations can be studied as new conditions. In addition, our work only consider global skeletons without fine-grained semantics. How to inject detailed semantics into the skeleton parts could also be a meaningful topic to study. More discussion can be found in appendix.

## 8 CONCLUSION

In this work, we propose to use skeletons as the structural condition for controllable generation. First, we construct a large-scale 3D mesh-skeleton paired dataset. We propose an effective mesh skeletonization method to generate mesh-aligned sparse skeletons with a high success rate. Based on the dataset, we present a skeletal conditioned multi-view generation pipeline. Coordinate color encoding and skeletal correlation module are proposed to realize efficient condition representation and injection. Furthermore, 3D meshes can be instantly reconstructed, followed by a refinement stage to achieve better texture quality. In summary, our work achieves controllable multi-view and 3D generation with arbitrary skeletons as conditions.

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
