# OpenReview forum: "SKDream: Controllable Multi-view and 3D Generation with Arbitrary Skeletons"
_ICLR.cc/2025/Conference — ICLR 2025 Conference Withdrawn Submission_

### Official Review · Reviewer_fn9E · 2024-10-26

**Soundness:** 3
**Presentation:** 2
**Contribution:** 3
**Rating:** 6
**Confidence:** 3

**Summary:**

This paper proposed a pipeline, SKDream, for skeleton-conditioned text to 3d generation and introduced a mesh-skeleton paired dataset, Objaverse-SK, for such a task.

**Strengths:**

The paper is well-written. The dataset construction and method pipeline are easy to follow. The proposed method is quite efficient in inference time ~20s for mesh reconstruction and ~60s for optional refinement.

**Weaknesses:**

1. I appreciate that the authors have created baselines based on SDEdit and SDEdit+COSAG. However, since the method illustrated in the pipeline shows that the skeleton still needs to be projected into 2D images, I wonder what if we just apply the 2D skeleton + text-conditioned image generation from ControlNet generation and then feed it into any single-view image-to-3d model, e.g. zero123? How much would performance gain be from the proposed multi-view images input to LRM?

2. Could the author provide a comprehensive experiment and discussion on comparison with SDS-based method e.g. DreamGaussian? For example, taking a skeleton + text conditioned image as input to 3D generation?

3. A user study would be helpful for judging the comparison of motion control and appearance control quality.

**Questions:**

Please see the weakness.

---

### Official Review · Reviewer_DvwN · 2024-10-28

**Soundness:** 3
**Presentation:** 3
**Contribution:** 1
**Rating:** 3
**Confidence:** 4

**Summary:**

The authors present a novel problem of controlled 3D generation, by conditioning from 3D skeleton representations. To facilitate the learning of this problem, a large-scale synthetic dataset is created with paired skeletion-mesh assets. The authors further study variants of skeleton encodings and correlation learning, and reach an effective model that produces satisfying quality and generalizability. Extensive experiments also show the results outperform baseline methods.

**Strengths:**

1. The proposed mesh skeletonization robustly applies to a wide range of mesh typologies and categories, the resulting dataset is of a high-quality and effectively facilitated the training process.
2. The presentation of the paper is clear, well-written and easy to follow.
3. Qualitative results show the proposed method produces reasonable results with the current skeleton representation and pattern, the generation process is also efficient and produces meshes in minutes.

**Weaknesses:**

1. The current method is based on a highly constrained environment, thus the major concern is its practical usability, in particular:

(i) Since the synthetic skeleton is created by MCF and graph partition, the resulting skeleton does not have a clear semantic meaning at each node, which is contradict to practical pipelines where each joints can be clearly defined and followed consistently. Therefore it's unclear how should the skeleton conditions be created in practice.
(ii) The work lacks demonstration of results from arbitrary manual created skeleton inputs, where skeletons with the same curve but varying node positions often exist. Experiments with manually created skeletons that have the same overall structure but different node placements and should be included and quantified to show how sensitive the current method is to these variations, which would provide valuable insights into the robustness of the method.
(iii) Specifying 3D skeleton conditions is more challenging than 2D skeletons with optionally relative depth, the authors should include a comparison with baselines that use 2D skeleton inputs, and leaving the depth ambiguity to be resolved by simply learning from the distribution of the data.

2. The current methods present limited insights in learning from skeleton correlations. Most modules are a simple adaption of the attention-based methods, while lacking in-depth analysis of its effects. The author should include more ablation studies on the effects attention-based modules, e.g. visualizations of the learned correlations, or showing the results after removing these modules.

3. The current evaluation also heavily biased on the mesh skeletonization, while containing limited evaluations and ablations in the generalization results. As above, the author may test on out-of-distribution skeleton types or evaluations on real-world datasets. This would help assess the practical applicability of the method beyond the synthetic dataset

Overall, I find this paper lacks enough technical contributions for acceptance.

**Questions:**

See the Weakness.

---

### Official Review · Reviewer_TrjV · 2024-10-28

**Soundness:** 3
**Presentation:** 3
**Contribution:** 3
**Rating:** 6
**Confidence:** 3

**Summary:**

This paper propose to use skeletons as the structural condition for controllable 3D generation. It first constructs a large-scale dataset containing mesh and skeleton pairs that cover diverse skeletal structures and develope a new pipeline for generating sparse skeletons from meshes. Then it proposes a multi-view 3D generation pipeline with arbitrary skeletal conditions, which includes coordinate color encoding for compact condition representation and skeletal correlation module for effective condition injection.

**Strengths:**

1. The Objaverse-SK dataset it builds is very useful, as it contains large-scale mesh-skeleton pair, which will be extremely useful for future research.

2. The proposed skeleton extraction pipeline is both efficient and with high success rate, outperforming existing methods by a large margin.

3. The Coordinate Color Encoding methodology is pretty novel to me, as it is an efficient way to represent projected 3D assets and can distinguish multi-view projections.

4. The skeletal correlation modeling is a rather simple but effective approach.

5. Extensive experiments and ablation studies have demonstrated the effectiveness of the proposed pipeline.

**Weaknesses:**

1. There exists grammar mistakes in the paper, please polish the writing.

2. In the skeletal extraction section, the author is missing a lot of details, me personally is quite curious about how to build the graph from curves. Maybe add a little more algorithmic details in the appendix part will be better.

3. The appearance refinement is not introduced clearly. I'm still wondering why we need to maintain a learnable texture map u and what's the motivation of this?

4. The quantitative experiments seems a little inadequate, can you add some more baselines for comparison?

**Questions:**

Please refer to previous part.

---

### Official Review · Reviewer_HTnw · 2024-11-01

**Soundness:** 2
**Presentation:** 2
**Contribution:** 2
**Rating:** 3
**Confidence:** 2

**Summary:**

The paper presents a mesh skeleton-based approach for 3D generation. However, the approach lacks novelty, essential experimental validation, and is not clearly presented.

**Strengths:**

The paper attempts to use skeletons for 3D generation and generates skeletons for the 3D dataset Objaverse.

**Weaknesses:**

The paper lacks novelty. Although the authors claim their pipeline for skeleton generation is novel, it simply combines two traditional approaches (MCF and DP) to extract skeletons from a mesh, without providing adequate motivation or context to demonstrate why this combination is innovative. Additionally, the paper lacks detail regarding the skeleton extraction process, such as the number of hyperparameters involved and guidance on setting these parameters. The comparison is limited to a learning-based method, RigNet, even though the proposed approach is not learning-based. The authors should expand the related work on mesh skeletonization and compare their method with more traditional mesh skeletonization techniques.

The multi-view and 3D generation pipeline also lack novely. For example, AnimatableDreamer already employs a skeleton-based 3D generation approach, contradicting the authors' claim that “their work is the first to achieve arbitrary skeletal-conditioned generation.” The use of a diffusion model in this approach is also not novel; while skeleton conditioning is applied, it is not unique, as AnimatableDreamer similarly conditions its diffusion model on skeletons during training.

The experimental validation is insufficient. In mesh skeletonization, the proposed approach is only compared to SDEdit, excluding other established methods such as those by Tagliasacchi et al. (2012) and Bærentzen & Rotenberg (2021), which are cited in the related work. For 3D generation, no experimental comparisons are made with state-of-the-art (SOTA) methods, even though numerous 3D generation techniques, including ProlificDreamer, MVDream, Farm3D, Text2Video-Zero, and AnimatableDreamer, should be included in the evaluation to better demonstarte the performance of the proposed approach.

**Questions:**

Both the skeleton generation and 3D generation approaches lack novelty, and essential experiments are missing. Further details are provided in the weaknesses.

---

### Official Review · Reviewer_qCaK · 2024-11-03

**Soundness:** 3
**Presentation:** 3
**Contribution:** 4
**Rating:** 8
**Confidence:** 4

**Summary:**

This paper investigates the task of multi-view images and 3D object generation conditioned on skeletal information. The main contributions include:
  - A robust method for extracting object skeletons based on curve skeleton representations, leading to the creation of the Objaverse-SK dataset with paired object-skeleton data
  - A novel skeleton representation method called Coordinate Color Encoding (CCE) that is more amenable to diffusion-based generative models
  - A Skeletal Correlation Modeling module for efficient Skeletal Guidance Injection into the MVDream backbone
  - Extensive experimental validation demonstrating the effectiveness of the overall approach. The necessity of individual components through comprehensive ablation studies

**Strengths:**

- The paper is well-written and easy to follow, with a clear presentation of motivation and contributions
- The research addresses a novel and significant problem of using skeletal information for efficient 3D generation guidance, which has been relatively unexplored
- The proposed CCE representation is well-justified and experimentally proven to provide better control compared to direct skeleton coordinate information
- The method demonstrates clear superiority over baseline approaches on evaluation datasets
- The evaluation metrics and methodologies are appropriately designed for the task
- The experimental section is comprehensive, effectively demonstrating the rationality and necessity of each proposed module

**Weaknesses:**

- Several details in the presentation require improvement:
  - The overall objective function should be explicitly stated. If not an end-to-end model, the supervision signals and objective functions for each stage should be clearly described
  - Regarding the skeletal condition representation experiments, qualitative results in Figure 8 should be aligned with Figure 9, showing comparisons between w/o CCE-D (Raw), CCE (color only), and CCE-D (color+depth) as conditions. The corresponding relationship between ablation experiments and notation in Section 6.1 should be clarified accordingly
- Otherwise, the work is relatively self-contained without significant issues

**Questions:**

- What specific data was used to train the Contrastive Object-Skeleton Alignment (COSA) adapter?
- Camera-Related Details Require Further Clarification:
  - How are camera parameters defined (angle-based or other representations)?
  - What is the form and dimension of camera pose embeddings in Skeletal Correlation Modeling (SCM)?
  - How are camera views represented during multi-view texture refinement? Are intrinsic and extrinsic parameters of a perspective camera model used for differentiable rendering texture optimization?
- The current implementation appears to treat all joints with full degrees of freedom. However, in reality, some skeletal joints for human (like elbows and knees) have constrained movement. What are the authors' future considerations regarding these anatomical constraints?
- Regarding Equation 2, please clarify:
  - Which variables are involved in the gradient computation of $\nabla L_{\text{COSA}}$?
  - What are the specific inputs required when using this gradient term as guidance?

---

### Note · Authors · 2024-11-15

**Comment:**

We sincerely thank all reviewers for their efforts and insightful reviews. We will further polish our work based on the valuable feedback.

**Withdrawal Confirmation:**

I have read and agree with the venue's withdrawal policy on behalf of myself and my co-authors.